# Sustainable poverty alleviation capacity construction of farmers in poverty-stricken areas under the background of rural revitalization

**Yujia Zhai**, **Lixiang Zhang***, **Anyi Xing**

School of Agricultural Economics and Rural Development, Renmin University of China, Beijing, China

* zlx001@ruc.edu.cn

## Abstract

Rural revitalization aims to consolidate the achievements of poverty alleviation. Furthermore, enhancing the sustainable poverty alleviation ability of farmers in poverty-stricken areas is a critical concern. Using samples of farmers in four regions of the Yunnan province in China, multidimensional poverty and sustainable poverty alleviation ability indexes were constructed. Furthermore, a multidimensional poverty double boundary method (A–F method) was used to identify multidimensional poverty, and binary logistic function in SPSS was used to perform a regression analysis between multidimensional poverty and sustainable poverty alleviation ability. Results showed that as the poverty dimension increased, poverty incidence rate in the four regions decreased, with middle poverty incidence rate being the highest. Ability poverty was then identified as the major type of poverty in the sample, with the group of poor farmers with a high dimension of poverty being the most affected. Therefore, based on the ability structure of sustainable farmers out of poverty, we analyze the prominent problem of the ability construction of sustainable poverty reduction and propose a path for promoting sustainable poverty reduction ability. Hence, we will effectively link poverty alleviation with rural vitalization.

## 1. Introduction

In 2020, China successfully achieved its poverty alleviation targets and eliminated absolute, regional, and overall poverty. China has entered a new stage of rural revitalization [1]. It is the aspiration and pursuit of the international community to consolidate achievements in poverty alleviation, solve the problem of relative poverty, help poor farmers develop their capacity for sustainable poverty alleviation, and promote the all-around development of human society and achieve common prosperity [2].

The universal phenomenon of poverty has been generally studied from the income perspective [3]; however, income provides limited information and does not encompass the full picture of poverty [4]. The connotation and definition of poverty are constantly changing and

**Funding:** This research was supported by the key research projects of philosophy and social sciences of the ministry of education in 2018, "Study on implementation path of rural vitalization strategy (no. 18JZD030)" and Hebei province innovation capacity improvement plan project, "Research on capacity building for science popularization (205576151D)".

**Competing interests:** The authors have declared that no competing interests exist.

extending. Since then, the definition of poverty has extended into multidimensional poverty [5] and capacity poverty [6]. Nobel Prize-winning economist Amartya Sen introduced the concept of "ability poverty," proposing that the root cause of poverty is not simply income deprivation but deprivation in the ability and opportunity to create income for the poor. Ability poverty entails that the poor are unable to obtain and enjoy a normal life [7]. In this regard, Amartya Sen proposed a poverty alleviation path based on the reconstruction of individual capacity, highlighting that equal rights for all residents followed by good education and health enable individuals to increase their income and potentially overcome poverty [8]. In the late 1980s, the concept of capacity building emerged. Concurrent with the emergence of "aid fatigue" in traditional development aid—along with people's dissatisfaction with the aid effect—capacity building emphasizes the development based on local resources, ownership and leadership, and importance of human resource construction. Hence, "assistance" should be transformed to "less dependence" and then "self-development" [9]. Among many factors affecting poverty alleviation, the ability of self-development of the poor is critical, and cultivating self-development is key in achieving poverty alleviation and not returning to poverty [10].

Based on the capability approach, Zhang used 2010–2018 panel data to track and analyze the overall decline of multidimensional poverty in rural China and determined that, among all indicators, durations of deprivation regarding education contribute the most [11]. Guo confirmed that poverty alleviation through labor transfer helps improve the livelihood capital of farmers in terms of enhancing their capacity, helping the poor escape poverty, and achieve stable employment [12]. Analyzing the relationship between the capability of the poor population and poverty based on Sen's ideas, Sang indicates that basic survival and development, productive capacity, and development capacity of the poor are key factors in achieving sustainable poverty eradication in the deep-poverty areas [13].

Owing to certain historical and geographical factors, poverty pervades in border areas in China wherein ethnic minorities gather. Among the 14 contiguous poverty-stricken areas in China, Yunnan province includes 91 counties in four districts, ranking first in the country. Hence, poverty alleviation is vital in areas like Yunnan province. Using multidimensional poverty theory, the empirical analysis incorporates rural family household survey data on four border areas in China: Western Yunnan, rocky desertification, Diqing, and Wumeng areas. Survey data from rural family households in the four areas of. Additionally, we identify multidimensional poverty characteristics of farmers and analyze the effect of sustainable poverty reduction ability of multidimensional poverty to help farmers improve sustainable poverty alleviation ability. We aim to effectively link our achievements in poverty alleviation to rural vitalization.

## 2. Identify multidimensional poverty

### 2.1 Research methods

To measure multidimensional poverty, Alkire and Foster proposed a relatively mature method of multidimensional poverty identification, summation, and measurement, namely, multidimensional poverty double boundary method (A–F method) to measure the multidimensional poverty of farmers. The multidimensional poverty index can help measure the intensity of poverty in poverty breadth, depth, and multiple dimensions [14]. T widely accepted in subsequent studies and applied to multidimensional measurement and policy research in countries worldwide [15]. This calculation method is established for poverty dimensions and indicators. Consequently, the critical value of each dimension index can be determined, with the setting based on micro data from poor families. We then build A poverty deprivation matrix and use this to identify multidimensional poverty [16]. The specific steps are outlined as follows.

**(1) Construct poverty deprivation matrix.** If the total number of sample families is N, and the poverty measurement dimension is D, then the entire sample observation matrix is as follows:

$$X = \begin{bmatrix} x_{11} & \cdots & x_{1d} \\ \vdots & \ddots & \vdots \\ x_{n1} & \cdots & x_{nd} \end{bmatrix},$$

where $x_{ij}$ represents the value of family i in dimension j, I = 1,2...N;J = 1,2..d.

$z_j$ represents the critical value of deprivation of dimension j, and $z_j$ is used to determined the poverty status of families in this dimension. Hence, the poverty deprivation matrix G is obtained from matrix X:

$$G = \begin{bmatrix} g_{11} & \cdots & g_{1d} \\ \vdots & \ddots & \vdots \\ g_{n1} & \cdots & g_{nd} \end{bmatrix},$$

Where $x_{ij} \leq z_j$, $g_{ij} = 1$, family i is deprived in dimension j, and this indicates that family i is poor in the dimension j. When $x_{ij} \geq z_j$, $g_{ij} = 0$, this indicates that family i has not been deprived in dimension j. Hence, household i is not poor in dimension j.

**(2) Identify multidimensional poverty.** By comparing the value of poverty deprivation $g_{ij}$ and the critical value of multidimensional poverty k, the multidimensional poverty deprivation share matrix can be obtained as follows:

$$c_{ij}(k) = \begin{cases} \sum_{j=1}^{d} g_{ij}, & \sum_{j=1}^{d} g_{ij} \geq k \\ 0 & \text{other} \end{cases},$$

Where $\sum_{j=1}^{d} g_{ij} \geq k$ means that the sum of the poverty of sample family i in at least k dimensions is $c_{ij}(k)$, K = 1, 2,...d.

To determine whether the sample families are in multidimensional poverty according to the value of the multidimensional poverty deprivation share, we calculated the poverty number of sample families in different dimensions. The multidimensional poverty deprivation matrix Q is then obtained as follows:

$$Q = \begin{bmatrix} q_{11} & \cdots & q_{1d} \\ \vdots & \ddots & \vdots \\ q_{n1} & \cdots & q_{nd} \end{bmatrix}, q_{ij}(k) = \begin{cases} 1, & c_{ij}(k) > 0 \\ 0 & \text{other} \end{cases}.$$

**(3) Poor aggregation.** Multidimensional poverty incidence rate H and average deprivation share A are used to calculate multidimensional poverty index M, and the most impoverished regions are determined by comparison. Hence, the calculation formulas are as follows:

$$H(k) = \sum_{i=1}^{n} q_{ij}(k)/n,$$

$$A(k) = \sum_{i=1}^{n} c_{ij}(k) / \sum_{i=1}^{n} q_{ij}(k) \times d,$$

$$M(k) = H(k) \times A(k).$$

**Table 1. Survey statistics.**

| Number | Cities | Counties | Households |
|---|---|---|---|
| Western Yunnan area | 5 | 9 | 668 |
| Rocky desertification area | 3 | 3 | 202 |
| Diqing area | 3 | 3 | 200 |
| Wumeng area | 3 | 3 | 208 |

**(4) Poverty decomposition.** Multidimensional poverty index can be decomposed according to dimension, region, time, and so on. Using the index, we measure the contribution of different dimensions or regions to the overall multidimensional poverty index to determine the main factors contributing to poverty. Hence, the calculation formula of decomposition by dimension is as follows:

$$P_j(k) = \frac{\sum_{i=1}^{n} g_{ij}/(n \times d)}{\sum_{i=1}^{n} \sum_{j=1}^{d} g_{ij}/(n \times d)} = \frac{\sum_{i=1}^{n} g_{ij}}{\sum_{i=1}^{n} \sum_{j=1}^{d} g_{ij}}.$$

## 2.2 Data sources and dimension index selection

To maintain representativeness and universality, 1–2 counties and 1–2 natural villages are selected in each prefecture (city) per region. In July 2018, we surveyed 1,278 poor households in 29 natural villages for 2 months (Table 1). Moreover, our research does not involve ethical issues, and minors were not included in the survey. Our research method informs the interviewees through the questionnaire. Based on the multidimensional poverty index of the Human Development Report 2013 and the MPI index (Alkire and Foster), this study employs the multidimensional poverty index in the analysis. Furthermore, this study uses mature multidimensional poverty theory combined with multiple poverty indicators from the outline (2011–2020) document spirit [17] and the information gathered after the interview and survey. Finally, a multidimensional poverty evaluation index is then constructed (Table 2).

**Table 2. Multidimensional poverty evaluation indicators.**

| Dimension | Indicator | Poverty line | Indicator description |
|---|---|---|---|
| Income | Per capita income | 2,300 yuan | If per capita income falls below the 2010 national poverty line, the value is 1. |
| Education | Educational background of household head | Junior high school | If the householder has less than junior high school education, the value is 1. |
| | Highest education | College | If no family member has a college degree or above, the value is 1. |
| Health | Disease | Major illness or disability | If a family member suffers from a major illness or disability, the value is 1. |
| | Medical insurance | N | If the family is not covered by NRCMS, the value is 1. |
| Life quality | Housing | Unstable house | If the family house is made from wood, grass, adobe, and other unstable materials, the value is 1. |
| | Drinking water | Unsafe water | If the family has no well or running water, the value is 1. |
| | Health facilities | No separate toilet | If the family does not have a separate toilet, the value is 1. |
| | Electric power | No electricity | If the family has no electricity at home, the value is 1. |
| | Fuel | Unclean energy | If the home does not use clean energy such as biogas, the value is 1. |
| | Household assets | Total assets | If the household owns less than three TV sets, washing machines, refrigerators, recorders, radios, computers, mobile phones, farm vehicles, tractors, and other farm machinery, the value is 1. |
| Social resources | Social support | N | If the family does not participate in production mutual aid organizations, agricultural business cooperatives, production skills training, and cash assistance, the value is 1. |
| | Poverty alleviation project | N | If the family does not participate in a poverty assistance program, the value is 1. |

### 2.3 Multidimensional poverty measurement and decomposition of farmers

According to the different dimensions of decomposition of four areas (Table 3), education in different dimensions (K = 3, K = 6, K = 10) is one of the main reasons why some families are poor. Another reason is social resources. As the number of dimensions increase, the proportion of education and social resources decreased and became evident. Families with higher dimensional poverty (K = 10) have more serious problems in aspects such as housing safety, drinking water safety, and sanitation. However, the contribution rate of electricity is the lowest among the 13 poverty indicators, indicating that the four areas have practically achieved access to electricity.

Owing to poor dimension decomposition, in families with different poverty levels, inadequate education causes families to experience *ability poverty*. Ability poverty also affects families' access to social resources and education, increases their contribution rate, and brings major diseases and disabilities to family members. Families are trapped in a vicious cycle of poverty that diminishes their quality of life in aspects such as housing and drinking water. On the one hand, when promoting equalization of basic public services in poverty-stricken areas, special attention should be paid to the intervention of human capital in these places. Poverty alleviation must nurture wisdom to enable individuals to obtain income. On the other hand, medical assistance should be enhanced to resolve deep poverty.

## 3. Multidimensional research on the sustainable poverty alleviation ability of poor farmers

### 3.1 Explanatory variables and their measurement indicators

Contemporary American scholar Michael Sheridan proposed a new theoretical model of antipoverty, which aimed to increase income while conducting asset construction for the poor, including tangible (e.g., capital, housing) and intangible assets (e.g., human capital, social capital, political capital) [18]. Sheridan highlighted that asset construction can realize the welfare effect of assets through education and training and loan subsidies for the poor. Subsequently, this can promote poverty alleviation and development. Domestic scholars believe that the development ability of farmers generally includes three aspects: internal forces (e.g., physical strength, mental strength, mental strength), external forces (e.g., natural resources and environment, social and economic conditions), and comprehensive abilities (e.g., labor skills, investment ability, communication ability) [19]. Therefore, the sustainable poverty alleviation capacity of farmers is constructed in this research, comprising physical quality, education level, production capacity, participation capacity, external information access ability, and social resource utilization ability. Explanatory variables comprised A total of 15 indicators, and the dependent variable consists of whether the investigated farmers experience multidimensional poverty. Low, medium, and high dimensions (K = 3, K = 6, K = 10) are utilized as dividing lines (i.e., when households are in poverty dimensions of 3 or above, 6 or above, and 10 or above, the value is assigned as 1; conversely, for the other dimensions, the value assigned is 0). Furthermore, the significance of the sustainable poverty alleviation capacity of farmers in different poverty dimensions is then studied. Changes in the sustainable poverty alleviation capacity of farmers in different poverty dimensions are compared and analyzed to investigate the loss of the sustainable poverty alleviation capacity of farmers as the degree of poverty deepens. Table 4 presents the specific variables and meanings.

### 3.2 Multidimensional loss of sustainable poverty alleviation capacity of poor households

The study adopts the binary logistic function in SPSS for regression analysis. Table 5 presents the analysis results.

Table 3. Multidimensional poverty index decomposition based on different dimensions (%).

| Dimension | Indicator | K = 3 | | | | K = 6 | | | | K = 10 | | | |
|---|---|---|---|---|---|---|---|---|---|---|---|---|---|
| | | Western Yunnan | Rocky desertification | Diqing | Wumeng | Western Yunnan | Rocky desertification | Diqing | Wumeng | Western Yunnan | Rocky desertification | Diqing | Wumeng |
| Income | Per capita income | 7.34 | 4.49 | 5.93 | 6.90 | 8.09 | 5.56 | 5.30 | 8.13 | 7.82 | 7.41 | 5.63 | 9.02 |
| Education | Educational background of household head | 11.7 | 13.1 | 10.9 | 12.5 | 10.2 | 11.8 | 8.83 | 11.0 | 8.89 | 7.41 | 9.86 | 9.77 |
| | Highest education | 16.9 | 15.1 | 14.8 | 15.3 | 12.4 | 12.9 | 10.6 | 12.9 | 8.89 | 9.26 | 8.45 | 9.77 |
| Health | Disease | 4.47 | 2.24 | 7.17 | 4.77 | 7.23 | 3.13 | 11.4 | 7.55 | 9.16 | 9.26 | 8.45 | 9.77 |
| | Medical insurance | 4.86 | 2.49 | 7.93 | 5.71 | 7.84 | 3.36 | 11.7 | 8.85 | 9.16 | 9.26 | 9.86 | 9.77 |
| Life quality | Housing | 2.85 | 1.83 | 6.21 | 4.77 | 4.65 | 2.43 | 9.79 | 7.26 | 7.28 | 7.41 | 9.86 | 9.77 |
| | Drinking water | 5.22 | 6.90 | 8.13 | 6.05 | 8.19 | 8.00 | 12.2 | 5.95 | 9.43 | 5.56 | 9.86 | 7.52 |
| | Health facilities | 7.22 | 9.39 | 6.41 | 4.43 | 7.84 | 9.73 | 6.26 | 4.35 | 7.55 | 9.26 | 8.45 | 2.26 |
| | Electric power | 0.33 | 1.16 | 0.86 | 1.11 | 0.40 | 0.58 | 0.80 | 0.87 | 0.27 | 0.00 | 1.41 | 0.75 |
| | Fuel | 13.7 | 15.7 | 15.0 | 11.1 | 11.6 | 13.4 | 11.7 | 9.72 | 9.16 | 9.26 | 9.86 | 9.77 |
| | Household assets | 4.11 | 6.23 | 0.48 | 2.73 | 4.70 | 7.88 | 0.64 | 3.77 | 6.47 | 7.41 | 2.82 | 3.01 |
| Social resources | Social support | 7.76 | 7.98 | 7.84 | 10.3 | 7.18 | 9.62 | 5.62 | 8.71 | 7.82 | 9.26 | 9.86 | 9.77 |
| | Poverty alleviation project | 13.5 | 13.5 | 8.32 | 14.4 | 9.71 | 11.6 | 5.14 | 10.9 | 8.09 | 9.26 | 5.63 | 9.02 |

**Table 4. Variable design and explanation.**

| Indicators | Variable | Variable assignment and meaning |
|---|---|---|
| Physical quality | Householder age | Age of householder. |
| | Labor | Number of household members in the labor force. |
| Education level | Householder educational background | Educational level of household head: illiterate, 1; primary school, 2; junior middle school, 3; senior high school or technical secondary school, 4; junior college or higher vocational, 5; bachelor's degree or above, 6. |
| | Number of students above junior high school | Number of family members with junior high school education or above. |
| Production capacity | Per capita arable land | Actual per capita arable land of family members. |
| | Production initiative | If "breeding," "food processing and operation," "wholesale and retail," "house construction," "agricultural leisure," and "other," the value assigned is 1. Otherwise, the value is 0. |
| Learning participation ability | Production of mutual aid organization | If the farmer participates in the production mutual aid organization, the value is 1. Otherwise, the value is 0. |
| | Skills training | If the farmer participates in production skill training, the value is 1. Otherwise, the value is 0. |
| | Assistance program | Participation in poverty assistance projects is assigned a value of 1. Otherwise, the value is 0. |
| | Business cooperation | If the farmer has a cooperative relationship with an agricultural business, the value is 1. Otherwise, the value is 0. |
| Information acquisition ability | Number of migrant workers | Number of family members who work for more than 6 months outside their home yearly. |
| | Working location | Value of "neighboring village" and "county seat" is 0, and the value of "provincial capital" and "outside province" is 1. |
| Ability to use social resources | Number of village cadres | Number of village officials in the family. |
| | Difficulty obtaining funds | Families have difficulty obtaining funds, degrees ranging from very difficult, 1; difficult, 2; somewhat difficult, 3; and no difficulty, 4. |
| | Cash assistance opportunities | Family access to cash assistance: rarely, 1; less, 2; generally, 3; more, 4; a lot 5. |

**Table 5. Multidimensional loss of sustainable poverty alleviation capacity of poor households.**

| Indicators | Dimension | K = 3 B | K = 3 Sig. | K = 6 B | K = 6 Sig. | K = 10 B | K = 10 Sig. |
|---|---|---|---|---|---|---|---|
| Physical quality | Householder age | −.059 | .058 | −.028 | .624 | .075 | .568 |
| | Labor | −.068 | .046 | −.038 | .476 | −.112 | .393 |
| Education level | Householder educational background | .027 | .878 | −.243 | .606 | −.164 | .440 |
| | Number of students above junior high school | .092 | .402 | .000 | .997 | .068 | .606 |
| Production capacity | Per capita arable land | .144 | .158 | .052 | .154 | −.107 | .370 |
| | Production initiative | −.774 | .019 | −.123 | .524 | .065 | .889 |
| Learning participation ability | Production of the mutual aid organization | −1.614 | .000 | −.623 | .000 | −1.988 | .009 |
| | Skills training | −.079 | .758 | −.108 | .427 | −.160 | .649 |
| | Assistance program | .025 | .922 | −.034 | .799 | .285 | .363 |
| | Business cooperation | −1.390 | .006 | −.542 | .020 | −.119 | .880 |
| Information acquisition ability | Number of migrant workers | .058 | .727 | .082 | .317 | .057 | .772 |
| | Working location | .392 | .213 | .199 | .189 | −.507 | .092 |
| Ability to use social resources | Number of village cadres | −.067 | .885 | .258 | .261 | −.862 | .385 |
| | Difficulty obtaining funds | −.292 | .025 | .019 | .757 | −.116 | .417 |
| | Cash assistance opportunities | −.106 | .485 | −.154 | .060 | −.620 | .037 |

For poor households with different poverty levels, various factors in sustainable poverty alleviation capacity play different roles and influence degrees. Regression results of low-dimensional poverty (k = 3) demonstrate that six variables—"age of household head," "labor force," "production enthusiasm," "production mutual aid organization," "business cooperation," and "difficulty in obtaining capital"—have significant impact on the occurrence of low-dimensional poverty, and all variable coefficients are negative. Thus, higher physical quality entails stronger productive, participation, and social resource utilization capacities. Additionally, the probability of multidimensional poverty is lower, and the family's participation in production of mutual aid organization and business cooperation have the most evident effect, represented by coefficients of −1.614 and −1.390, respectively. Furthermore, the coefficient of farmers' enthusiasm for production and difficulty in obtaining funds are −0.774 and −0.292, respectively. At the 0.1 level, the age of the household head and the number of household members in the labor force are significant, with coefficients of −0.059 and −0.068, respectively. Relying solely on cultivated land income to improve their family role limits the household members' capabilities. Hence, mutual aid organizations involved in production and business cooperation can effectively help farmers develop production skills and receive help. Cultivated land outside small production and operation activities can also enable farmers to increase their income and improve their living standards. In time, farmers' abilities in other aspects will also develop. Therefore, these factors are negatively correlated with the occurrence of multidimensional poverty. Considering the results of the development capacity of rural households in poverty (k = 6), significant variables are reduced to three with negative coefficients—"mutual aid organization," "business cooperation," and "cash assistance opportunities"—with coefficient values also decreasing to −0.623, −0.542, and −0.154, respectively. Both physical quality and production capacity can no longer explain the occurrence probability of multidimensional poverty. Furthermore, the influence of participation ability and social resource utilization ability on the occurrence probability of multidimensional poverty has been reduced. As research objects are high-dimensional (k = 10) poor farmers, the content changes. The influential factor to farmers' access to information is "working location"; essentially, rural households have household heads working in the provincial capital of Fujian and even as migrant workers. Hence, these households are more open to accepting outside information and have stronger processing power. Therefore, this factor is negatively correlated with the occurrence of multidimensional poverty, and the correlation coefficient is −0.507. Additionally, the ability to participate and use social resources remain effective. The significant variables are "production of mutual aid organization" and "cash assistance opportunity," with variable coefficients of −1.988 and −0.620, respectively.

Analysis results reveal that in the participation ability and social resource utilization ability of sustainable farmers out of poverty for low, medium, and high dimensions, poverty incidence has significant impact on meaning. Hence, participating in the social resource utilization ability and strengthening it can help farmers enhance communication, promote resource sharing, understand the market, control technology, and support financing. In turn, these improvements effectively reduce poverty incidence. Overall, the present establishes six development capacities and 15 factors. With deepening of poverty, some of the farmers' development capacities are lost. For low-dimensional poor peasant households, four capacities and six factors are active, namely, physical quality, production capacity, participation capacity, and social resource utilization capacity. For poor peasant households in group d who only participate in the social resource utilization ability and ability of two, three factors are significant. Meanwhile, for high-dimensional poor peasant households, in addition to participating in the social resource utilization ability and ability, the effect of "working location" in the access to information ability is enhanced. Conversely, degree of poverty, physical quality, and

production capacity of the existing economic development utility are reduced. Owing to the deep-poverty situation, becoming a migrant worker has become one of the main methods to improve economic conditions among households. Furthermore, the influence level of education is always not significant because in the traditional, low-skill labor market, farmers' physical quality and production capacity are reduced. Additionally, the more educated lose their comparative advantage.

## 4. Sustainable poverty alleviation capacity building for rural households

Based on a combination of tangible and intangible resources owned by individuals and families, this study selects six sustainable poverty alleviation capabilities of farmers—physical quality, education level, production capacity, participation capacity, information acquisition capacity, and social resource utilization capacity—to investigate the capacity loss of poverty groups in different dimensions. First, we observed that the level of education in low, medium, and high dimensions among farmers does not retain its original economic development effectiveness. Low-dimensional poverty groups were significant in the statistical analysis ability of physical quality and production capacity. Moreover, these groups participate in the social resource utilization ability and ability. In the group, poor peasant households lose their physical quality and production capacity and only participate in social resource utilization ability and ability to influence. Additionally, the ability to acquire information gradually emerges in poverty-stricken peasant households. Loss of sustainable poverty alleviation capacity of farmers considerably impacts the middle- and high-dimension groups, affecting these families' accumulation of wealth and improvement of life quality. Our study proposes suggestions on the sustainable poverty alleviation capacity construction of multidimensional poor farmers centering on human capital investment.

First, investment in rural education must continually increase to reduce school-aged teenagers' dropout rate. Populations in poor villages are mainly characterized by short schooling years, middle and old age, and high proportion of illiteracy in the minority population. To improve the overall quality of the future rural population, basic education must first be addressed. Although the centralization of schools in villages and towns has reduced the dropout rate of school-aged children in poor villages, teenagers' dropout rate remains high. Given various reasons, children of poor families are out of school and at home. Thus, they neither have the ability to go out for work nor engage in high value-added production activities at home. Hence, the government should establish special funds to subsidize or incentivize children of poor peasant households to complete compulsory education and receive appropriate professional skills training in the process of financial budget or social poverty alleviation fundraising.

Second, the learning and production capacity of multidimensional poor farmers must be improved. Additionally, the skills training (and content of training) of poor farmers must be improved. In the development of the poverty population problem, low efficiency of resource use remains prominent. Additionally, low individual basic ability is related to lack of message exchange with the outside world. Production and living areas of poor peasant households can rapidly improve through agricultural skills and popular science knowledge. Moreover, the lack of health knowledge has greatly restricted the development of peasant households. The enhancement of farmers' basic skills is a long-term task, and short-term training can expand the objectives of poor farmers and help them actively pursue development opportunities. Additionally, we will continue expanding the number of students from poverty-stricken areas enrolled in colleges and technical secondary schools, support a number of specialties with local

characteristics and advantages, and cultivate technical and skilled personnel needed for local industrial development.

Finally, multidimensional poor peasant households must be provided with aid to develop social capital and facilitate the interventional role of professional social workers. Additionally, the construction of community mutual aid organizations must be strengthened, ability of cooperation and mutual aid within the organizations must be improved, innovation and pioneering ability of community organizers must be enhanced, development of farmers in the community must be promoted, farmers' mindset of "waiting, relying on, and wanting" must be changed, and farmers' sense of social participation and self-reliance must be developed. Village-based communities have formed an organizational structure with high adaptability through a democratic election system. Practice has proven that community organizations have played positive roles in poverty alleviation work. However, considering the perspective of improving organizational efficiency, assistance and mutual assistance within the community need to be further strengthened. Particularly, such assistance includes improving the information and resource-sharing mechanism within the organization, creating suitable development platforms and projects, and allowing high-income farmers to lead low-income farmers and those with employment channels to help unemployed farmers. These activities ultimately aim to accelerate the construction of the sustainable poverty alleviation capacity of poor farmers.

## Supporting information

**S1 File. Sample questionnaire.** This is a sample of the questionnaire.
(DOC)

**S2 File. Questionnaire statistics.** This is the statistics from the questionnaire.
(XLSX)

## Author Contributions

**Conceptualization:** Yujia Zhai, Lixiang Zhang.

**Data curation:** Yujia Zhai, Lixiang Zhang, Anyi Xing.

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
