## [Decision Letter · Decision Letter 0]

29 Aug 2022

PONE-D-22-10030Research on sustainable poverty alleviation capacity construction of farmers in poor areas under the background of rural revitalization

Dear Dr. Zhai,

Thank you for submitting your manuscript to PLOS ONE. After careful consideration, we feel that it has merit but does not fully meet PLOS ONE’s publication criteria as it currently stands. Therefore, we invite you to submit a revised version of the manuscript that addresses the points raised during the review process.

Please address the following issues:

Method(s) is inadequately written in the abstract.Cite the literature consistently and give credits to authors to support your stories in the introduction instead of citing the studies [9-11) together at the end of a sentence.Table heading seems to be orphan. Write them correctly.Check grammar and typos throughout the manuscript.Follow journal's format and address reviewers' comments.

Kind regards,

Md Nazmul Huda, PhD

Academic Editor

PLOS ONE

Journal Requirements:

2. Thank you for your ethics statement reading "The research does not involve ethical issues. The research did not involve minors. The research method of this manuscript is to inform the interviewees in the form of questionnaire." At this time, we request that you please confirm that the data was collected and analyzed anonymously, and that the authors did not collected any identifying information from the participants. Thank you for your attention to this request.

Reviewers' comments:

Reviewer's Responses to Questions

**Comments to the Author**

1. Is the manuscript technically sound, and do the data support the conclusions?

Reviewer #1: Yes

Reviewer #2: Yes

2. Has the statistical analysis been performed appropriately and rigorously? 

Reviewer #1: Yes

Reviewer #2: Yes

3. Have the authors made all data underlying the findings in their manuscript fully available?

Reviewer #1: Yes

Reviewer #2: Yes

4. Is the manuscript presented in an intelligible fashion and written in standard English?

Reviewer #1: Yes

Reviewer #2: Yes

5. Review Comments to the Author

Reviewer #1: The work presented in this paper is interesting, in particular the data and the analysis which have been made on the data and the outcome which specifies factors affecting the poverty in rural areas. Nice work that show good analytical expertise.

The paper is not following the journal format, such as the titles, the bold face font, the spacing, etc making it difficult to follow in terms of where sections or paragraphs start etc.

Reviewer #2: The manuscript includes information to support its conclusions. The conclusions were correctly drawn by the author using an appropriate sample size and the provided data.

The manuscript does, however, require a few corrections to the grammar and punctuation.

6. PLOS authors have the option to publish the peer review history of their article (what does this mean?). If published, this will include your full peer review and any attached files.

Reviewer #1: No

Reviewer #2: **Yes: **Nazli Mohammad

---

## [Author Response · Author response to Decision Letter 0]

13 Oct 2022

Response to Reviewers

1. Method(s) is inadequately written in the abstract.

Response 1:In the abstract, the description of the method is added, and the modification is in line 9-13. In addition, the method and process descriptions are added in the paper, which are detailed in lines 70-77, 88-93, 98-103, and 112-114.

2. Cite the literature consistently and give credits to authors to support your stories in the introduction instead of citing the studies [9-11] together at the end of a sentence.

Response 2:The literature review is supplemented and modified, such as the specific research background, multidimensional poverty research perspective, empirical research in China, etc., in line 24-32, line 36-39, and line 44-57, respectively.

3. Table heading seems to be orphan. Write them correctly.

Response 3:The table has been examined and is supplemented with a description of Table 3 in lines 132-136.

4. Check grammar and typos throughout the manuscript.

Response 4: Grammatical and typos errors were corrected and the manuscript was polished by a professional body. Please see the attachment for proof.

5. Follow journal's format and address reviewers' comments.

Response 5: It has been revised according to the format requirements of the journal and the comments of reviewers.

Response to Editors

Response 1: It has been revised to comply with the style requirements and naming requirements of the journal.

2. Thank you for your ethics statement reading "The research does not involve ethical issues. The research did not involve minors. The research method of this manuscript is to inform the interviewees in the form of questionnaire." At this time, we request that you please confirm that the data was collected and analyzed anonymously, and that the authors did not collected any identifying information from the participants. Thank you for your attention to this request.

Response 2: We have a supplementary statement on lines 118-120.

Response 3: We checked the funding information.

Response 4: We supplement and modify the references, and the modification is in line 314-425.

---

## [Editor Report · Decision Letter 1]

14 Oct 2022

Sustainable poverty alleviation capacity construction of farmers in poverty-stricken areas under the background of rural revitalization

PONE-D-22-10030R1

Dear Dr. Yujia Zhai,

We’re pleased to inform you that your manuscript has been judged scientifically suitable for publication and will be formally accepted for publication once it meets all outstanding technical requirements.

Kind regards,

Md. Nazmul Huda, PhD

Academic Editor

PLOS ONE

---

## [Editor Report · Acceptance letter]

7 Dec 2022

PONE-D-22-10030R1 

Sustainable poverty alleviation capacity construction of farmers in poverty-stricken areas under the background of rural revitalization 

Dear Dr. Zhai:

I'm pleased to inform you that your manuscript has been deemed suitable for publication in PLOS ONE. Congratulations! Your manuscript is now with our production department. 

Kind regards, 

on behalf of

Dr. Md. Nazmul Huda 

Academic Editor

PLOS ONE